# *Regulus* infers signed regulatory relations from few samples' information using discretization and likelihood constraints

**Marine Louarn**[1,2¤a], **Guillaume Collet**[1], **Ève Barré**[1], **Thierry Fest**[2,3], **Olivier Dameron**[1], **Anne Siegel**[1], **Fabrice Chatonnet**[2,3¤b]*

**1** Univ Rennes, CNRS, Inria, IRISA - UMR 6074, Rennes, France, **2** UMR_S 1236, Université Rennes 1, INSERM, Etablissement Français du Sang, Rennes, France, **3** Laboratoire d'Hématologie, Pôle de Biologie, CHU de Rennes, Rennes, France

¤a Current address: Michael Smith Laboratories, University of British Columbia, Vancouver, British Columbia, Canada
¤b Current address: U1111 CIRI, Lymphoma ImmunoBiology team, INSERM, Université de Lyon, Oullins, France
* fabrice.chatonnet@ens-lyon.org

**Data Availability Statement:** The data underlying this article are available in the article and in its online supplementary material. Regulus software and source code are available on gitlab at: https:// gitlab.com/teamDyliss/regulus. ClassFactorY

## Abstract

### Motivation

Transcriptional regulation is performed by transcription factors (TF) binding to DNA in context-dependent regulatory regions and determines the activation or inhibition of gene expression. Current methods of transcriptional regulatory circuits inference, based on one or all of TF, regions and genes activity measurements require a large number of samples for ranking the candidate TF-gene regulation relations and rarely predict whether they are activations or inhibitions.

We hypothesize that transcriptional regulatory circuits can be inferred from fewer samples by (1) fully integrating information on TF binding, gene expression and regulatory regions accessibility, (2) reducing data complexity and (3) using biology-based likelihood constraints to determine the global consistency between a candidate TF-gene relation and patterns of genes expressions and region activations, as well as qualify regulations as activations or inhibitions.

### Results

We introduce *Regulus*, a method which computes TF-gene relations from gene expressions, regulatory region activities and TF binding sites data, together with the genomic locations of all entities. After aggregating gene expressions and region activities into patterns, data are integrated into a RDF (Resource Description Framework) endpoint. A dedicated SPARQL (SPARQL Protocol and RDF Query Language) query retrieves all potential relations between expressed TF and genes involving active regulatory regions. These TF-region-gene relations are then filtered using biological likelihood constraints allowing to qualify them as activation or inhibition. *Regulus* provides signed relations consistent with public databases and, when applied to biological data, identifies both known and potential new

software and source code are available on gitlab at: https://gitlab.com/teamDyliss/ClassFactorY. Regulatory Circuits datasets and networks, FANTOM5 and GTex data used for testing and validation are available in the supplementary data of marbach2016tissue at http://www2.unil.ch/cbg/regulatorycircuits/Supplementary_data.zip and http://www2.unil.ch/cbg/regulatorycircuits/FANTOM5_individual_networks.tar. B cells datasets are publicly available on GEO, as datasets GSE136988 (RNA-seq for PB) and GSE190458 (all other data).

**Funding:** ML was financed by the "Médecine Numérique" joint PhD program from INRIA & INSERM. Data acquisition on B cells subsets was funded by an internal grant from the Hematology Laboratory, Pôle de Biologie, Centre Hospitalier Universitaire de Rennes, Rennes, France to TF. The funders had no role in study design, data collection and analysis, decision to publish, or preparation of the manuscript.

regulators. *Regulus* is devoted to context-specific transcriptional circuits inference in human settings where samples are scarce and cell populations are closely related, using discretization into patterns and likelihood reasoning to decipher the most robust regulatory relations.

## Author summary

Gene expression regulation is based on the activity of specialized regulatory proteins called transcription factors (TFs) which can bind DNA at specific sequences. Understanding the regulatory relations between TFs and genes in humans is fundamental in personalized clinical settings, to better decipher the pathological mechanisms and to identify new therapeutic solutions. However, finding the main regulators of such systems is usually difficult, due to the scarcity of available samples and the biological closeness of the studied cell types. To overcome these issues, we introduce a new tool called *Regulus*. We use information from genes and TFs expression, regulatory regions activity and TF binding sites occurrences to compute TF-gene relations. We then apply a likelihood reasoning step, based on the biological knowledge of transcriptional regulation mechanisms, to select the most probable relations and assign them a function as activation or inhibition. Finally, we reduce the potential TFs list by a specificity / coverage filter and we annotate it according to existing literature. By testing *Regulus* on large-scale biological datasets, each describing four biological contexts, we show that this tool is able to i) identify both known and undescribed regulators consistent with all the gene expression and region accessibility constraints in each biological context, ii) include low expressed genes in its relations and iii) considerably limit the space of putative TF-gene relations.

## Introduction

### Finding context-specific transcriptional regulators in few samples

Gene expression regulation a also known as transcriptional regulation) is a major field of investigation in life science. It allows a better understanding of major processes such as cell differentiation, cell identity and cell transformation [1].

Transcriptional regulators are specialized proteins called transcription factors (TFs) which bind DNA in regulatory regions at specific binding sites. This results in activation or inhibition of target gene expressions. Regulatory regions are located in non-coding DNA [2] and their accessibility, dependent on 3D chromatin conformation, constrains TF binding to DNA [3]. A regulatory region in a "closed" conformation will therefore limit any potential regulation for a TF having a binding site in this region. There are strong inter-dependencies between gene expression, TF availability and region accessibility, as i) TF are themselves encoded by genes subject to transcriptional regulation, ii) TF can modify chromatin accessibility and iii) accessibility can be regulated by gene expression. Therefore, biological constraints strongly affect the observation of TF-gene regulation relations.

Consequently, gene regulation is extremely context-dependent [4–6]: pathological processes such as cancer can disturb the transcriptional regulatory circuits by modifying regulatory regions accessibility or location or by modifying TF fixation abilities [7]. Information about regulatory regions and TF expression is therefore essential to build reliable context-dependent regulatory interactions and predict the pathological effects of genome perturbations, paving the way to personalized treatments. This high context sensitivity is a major

challenge, as obtaining large quantities of data required by the classical analysis methods is complicated in most experimental designs, where only a limited number of samples (in the range of 10–20 samples representing 4 to 8 cell populations) is available.

## Use case: Regulation driving B cells differentiation

As an illustration, let us consider the biological case study of differentiation of human B cells into antibody producing cells. This process involves several transitions between closely related cell types, which are finely regulated by genetic circuits [8]. Their deregulation can lead to immunodeficiencies, autoimmune diseases and hematological malignancies. Very few large scale studies have been performed to understand B cell normal and pathological differentiation. In [9], the author infer regulatory relations based on statistical analysis of TF and gene co-expressionbut do not take the chromatin context (TF binding sites and region accessibility) into account. In [10], only one type of B cells is described compared to other cell types and specific B-cells subsets can therefore not be differentiated. Other circuits are only built with a limited set of regulators [11] or based on review of the literature. For example, [8] describes two main sets of opposed regulators: BACH2, PAX5 and BCL6 inhibit terminal differentiation, while IRF4, PRDM1 and XBP1 induce it. Therefore a more complete characterization of B cells transcriptional regulatory circuits is required to better understand the hijacking of the normal differentiation process by cancerous cells.

We reviewed the main current transcriptional regulatory circuits inference methods for their ability to generate a reliable model of normal and pathological B cell differentiation, using the following criteria: (1) use of genes expression, TFs binding and regulatory regions activity data, (2) applicable to limited numbers of human samples, (3) able to predict inhibitions and activations and (4) reproducible and reusable on new datasets. The summary presented in S1 Table shows that most of these methods are not applicable to limited numbers of closely related human samples, do not use the context biological constraints between genes, TF and regions and rarely provide the activation or inhibition function of candidate TF-gene relations. These issues limit the possibility to use these methods to build a transcriptional regulatory landscape for B cell differentiation.

## Working hypothesis of TF-gene relations inference with few samples

Working with a limited number of samples imposes particularly strong limitations about the methods. It precludes methods based on statistical / correlation analysis or machine learning modelling which require a large number of values for statistical power, reliability [12] or training. In such a situation, we introduce an alternative strategy based on the identification of regulatory processes which consistently link paired measurements of gene and TF expressions and regulatory regions accessibility among the different studied samples to identify the most robust regulators of a patho-physiological context. Our hypothesis is that comparing the different values of an entity activity among samples through a discretization procedure lead to TF, target gene and region activity profiles (which we call "patterns" throughout this article) which implicitly contain signals of regulations. More precisely, our method assumes that when a TF-gene relation occurs for each studied situation, the discretized profiles of the TF expression, its target gene expression and the region accessibility comply with a very specific set of constraints. Therefore, TF-gene relations can be deciphered by identifying all the triples of TF expression, gene expressions and region activity patterns satisfying all the likelihood constraints implied by the existence of a regulation.

### Semantic Web technologies as a regulatory circuits building tool

The generation of (1) the TF / gene expression and region activity patterns and (2) the family of possible TF-gene regulation candidates which could satisfy the likelihood biological constraints, requires a strict and reproducible procedure for the integration of data and knowledge on TF, genes and chromatin. To that goal, a state-of-the-art framework for integration and reasoning over large-scale data is based on Semantic Web technologies [13, 14]. The Semantic Web has long been recognized as a relevant framework in life sciences [15], and has been widely adopted in this field [16–18]. In this framework, integration is achieved by Resource Description Framework (RDF), and querying by SPARQL (SPARQL Protocol and RDF Query Language).

In previous works [19, 20], we applied Semantic Web technologies to the *Regulatory Circuits* project [10] to generate a global RDF dataset exposed as a SPARQL endpoint. This allowed to recompute the TF-gene relations published by the *Regulatory Circuits* project on all or a subset of the original datasets, to provide users with the published TF-gene relation scores and to enrich the dataset with relations between samples, tissues and all metadata [20]. This successful strategy therefore provides a backbone for testing the relevance of TF-gene relations inference based on discretization into patterns and biological constraints filtering.

### Objective

In this article, we introduce *Regulus*, an inference tool to identify the most robust TF-genes regulation relations and qualify them as activation or inhibition. *Regulus* has been developed to be stringent and to limit the space of the candidate TF-genes relations, highlighting the candidate relations which are the most likely to occur. Its main principles are to (1) take into account regulatory factors (TFs and regions) activities, (2) discretize the activities into patterns, (3) produce signed circuits inferred by testing biological constraints likelihood and (4) be easily reusable and applicable to many datasets. By applying *Regulus* to various biological datasets, we show that it can describe regulatory circuits with validated signed relations, it is able to identify known regulators of a specific biological process and it provides a list of new candidate regulators.

## Results and applications

### The *Regulus* tool

We designed *Regulus*, a transcriptional regulatory circuits inference tool dedicated to the analysis of datasets with few and biologically-close samples.

As stated in Introduction, statistics, correlation or machine learning can not be fully leveraged in this setting. Thus, *Regulus* uses the concept of entity pattern to describe and discretize the data and biological likelihood constraints on discrete pattern values from different entities to filter the TF-gene relations. The tool relies on Semantic Web to integrate expression and epigenetic data. Fig 1 presents the different steps of the *Regulus* pipeline, illustrated by a "toy example" in S1 Fig. In a given context with few samples describing several cell states, it allows to compute the set of TF-gene relations which provide a consistent explanation of patterns of genes expressions, regulatory regions activities, TF binding sites and genomic coordinates of all features. In this study, we apply *Regulus* only to human data, but it can be used with any species and reference genome, as long as users have access to genomic coordinates for the three required features: genes, regions and TF binding sites.

**Pre-processing data for efficient integration: Patterns as descriptions of data dynamics.** The preprocessing steps (blue boxes and step (1) in the Fig 1B) consist in (1) finding

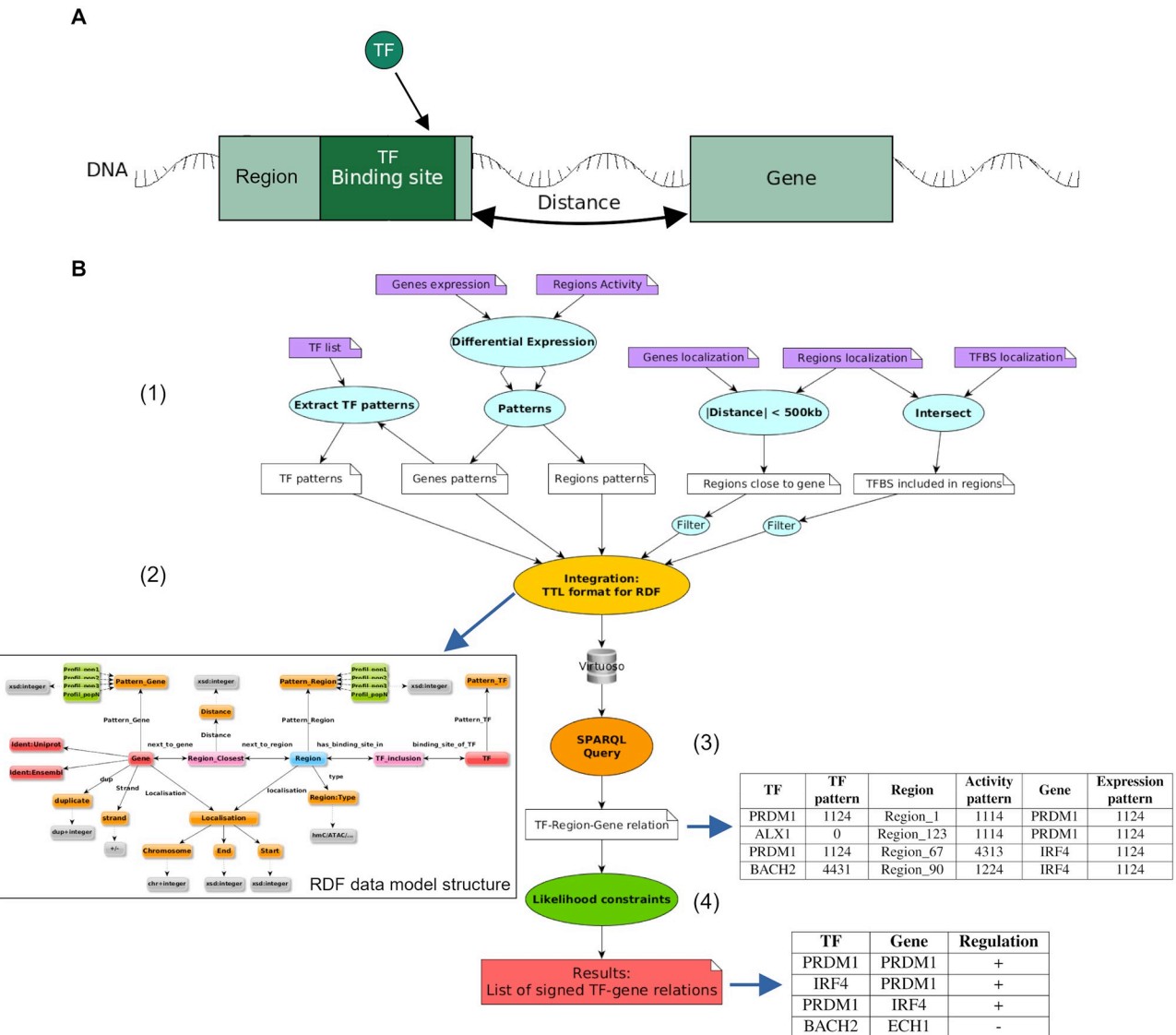

**Fig 1. The *Regulus* pipeline.** (A) Scheme introducing the biological relationships between the different entities used to compute the transcriptional regulatory circuits. (B) Representation of the different steps of the pipeline. (1) Input data (purple boxes) are pre-processed (blue bubbles) according to the biology, expliciting the relations between the entities—inclusion of the transcription factors binding sites (TFBS) in the regions (filtered to keep a single occurrence), distance between regions and genes (filtered to be less than 500 kb)—or reducing the data size and complexity by creating activity patterns for the different entities. (2) Integration and creation of a RDF graph formalizing the relations between all the entities (yellow bubble), generating a data model structure. (3) Dedicated SPARQL query to extract all the relations following the requirements introduced by Regulus (orange bubble) resulting in a global unsigned regulatory network expliciting relations as TF-region-gene triples and their respective patterns. (4) Application of biological likelihood constraints (green bubble) results in a signed and filtered regulatory circuit consisting in unique TF-gene relations (pink box).

relations between genes and regions by computing base pair distances between them, (2) finding the inclusion of TF binding sites into regions, (3) transforming individual entities (genes, TF and regulatory regions) activities (expressions and read densities) into patterns. The relevant entities for building a regulatory circuit are those which activity varies between the compared cell populations. Our circuit inference method is based on the common assumption that genes sharing a common expression dynamics are regulated by a common set of regulators [21]. For each entity, *Regulus* therefore aggregates activity dynamics measured in *n* different

cell populations into patterns, as a *n*-tuple. Each element of this pattern represents a cell population and its value represents the relative activity of this entity in the considered cell population compared to all other populations, coded on four levels (1 to 4, see Methods subsection *Pre-processing* and S2 Fig for details, illustrated in S1 Fig top panel). Each computed pattern therefore regroups entities exhibiting similar activity variations between cell populations regardless of their respective absolute activity levels.

**Data integration in a structure that can be browsed and queried.**   Pre-processed data are then integrated using Semantic Web technologies (yellow box and step (2) in Fig 1B). The data model structure after integration is illustrated in Fig 1B and S3 Fig, and can be seen as a representation of the interactions between the data, where the entities are linked with each other by explicit relationships. To retrieve TF-gene relations, the strictly necessary entities are: genes, TFs and regions with their respective patterns, as well as the reified relations Region_closest and TF_inclusion (see Methods subsection *Data graph for integration and query*, illustrated in S1 Fig middle panel). Other entities or properties can be kept to refine the results if needed.

Integrated data can then be interrogated to identify the candidate relations between entities that match a given set of rules: the regions must be at most at 500 kb of the gene, the TF must be expressed and have a binding site in the region (orange box and step (3) of Fig 1B). From the data structure, the corresponding SPARQL query (S4 Fig) is generated as follows: (1) starting from the node *Gene* with a specific expression pattern, (2) all *Regions* which are connected to the *Gene* by a *Region_closest* relationship are identified, (3) these *Regions* are associated to *TF* through *TF_Inclusion*, representing the presence of TF binding sites into regions. Along the way, the TFs, regions and genes patterns are retrieved, as they are required in the next step. The output of this step is a TF-region-gene relations table (see (3) in Fig 1B).

**Filtering and sign attribution by biological likelihood constraints.**   Query results are then refined (green box and step (4) of Fig 1B) to prioritize the biologically most likely TF-region-gene candidate relations and to infer a regulation sign (i.e. activation or inhibition) for these relations.

We use the following biological principles for gene regulation: **(1)** the maximum effect on gene expression is obtained when the TF is at its highest expression level: for an activation this implies that if the TF is highly expressed so must be the gene, for an inhibition the higher the TF, the lower the gene's expression. **(2)** The more accessible the region, the higher is the impact of the TF on the gene level; if the region looses its accessibility, the weight of the TF decreases: a higher TF expression level is needed to get a similar effect on gene expression.

These principles are modeled by a stringent formula corresponding to the "perfect" situation where the {TF expression pattern, gene expression pattern, region activity pattern} triplet gives an unambiguous regulation over all the considered cell types (Eq 1 in Methods subsection *Regulatory likelihood constraints*, illustrated in S1 Fig bottom panel). To take into account the situations where either the gene expression or its modulation by the region accessibility deviate from the "perfect" state, we also introduce a tunable relaxing parameter (Eqs 2 and 3 in the above-mentioned Methods subsection, respectively). These equations are then translated into a table (S5 Fig) and constraints are implemented by using a custom *Python* script.

After applying the likelihood constraints, the result is a quadruple between the gene, the neighboring regulatory region, the TF with a binding site in the region and the signed potential regulation on the gene expression. These relations can be merged into unique TF-gene relations when consistent relations involving the same TF and gene are found through different regions. The final output of the process is a signed TF-gene interaction table (see step (4) in Fig 1B).

## Application to *FANTOM5* data: Validation of the basic principles

One aim of *Regulus* is to compute regulatory circuits on a limited number of samples and cell populations. *Regulus* was therefore run on four limited datasets extracted from *FANTOM5* and used in [10], each containing four cell populations, chosen to be either of similar (comparable organ) or dissimilar (widely different localization) origin. Details about the chosen *FANTOM5* subsets are shown in Fig 2A. Data comprise 16,888 genes expressions, 43,012 regulatory region activities and 124,358,159 TF binding sites.

**Regulus generates context-dependent circuits which include low expressed genes.   Circuits topology is refined by applying biological likelihood constraints**. Fig 2B presents the number of relations obtained by *Regulus* for these four *FANTOM5* datasets. Just after the query and before taking the biological constraints into account, the same number of relations (3,005,934 TF-region-gene or 1,869,854 TF-gene) is generated for the four *FANTOM5* datasets. This is due to the fact that the exact same information about the TF binding sites and regions is used, leading to identical relations.

As seen in Fig 2B the constraints step allows to generate circuits which are different in size and quality. They represent about 10% of the possible relations predicted by the query, underlining the filtering power of using biological likelihood constraints. These constraints also modify the circuit topology by changing the distribution of the number of potential regulators per gene. As shown in Fig 2C and 2D, whereas in the total network genes can have up to 400 potential regulators with a mod of the distribution around 100, the filtered circuits have a more biologically realistic number of regulators per gene, ranging from 1 to 100 with a mode of 2 or 3. Moreover, this filtering allows to prioritize relations which are relevant to the biological context. Relations specific to each *FANTOM5* dataset are indeed more numerous than common relations (Fig 2E). These latter could be related to common biological processes necessary for basic cellular functions.

Circuits computed on dissimilar sets of cell populations are slightly smaller than the ones computed with similar subsets. This lower number of relations may be explained by the increased lack of likelihood in TF-gene regulations across widely different cell populations. They also include a higher proportion of negative relations, which may be easier to identify when comparing unrelated cell populations.

**Inclusion of low-expressed genes**. Gene inclusion in our circuits is validated with the same *Roadmap Epigenomics* RNA-seq datasets used in [10]. As seen in Fig 2F, highly and medially expressed genes are recovered at high levels (> 90% for each category). Low-expressed genes are also significantly included in *Regulus* circuits, with a retrieval rate ranging from 36.75% for similar cell types to 52% for dissimilar subsets.

**Impact of the likelihood constraints**. To test the impact of the likelihood constraints, different values are tested in S6 Fig (see Methods). We observe that combining a deviation of 1 from the optimal score and the region constraint gives a moderate increase in the number of TF-gene relations (1.32 to 2.36 times bigger). Meanwhile, the ratio of activations over inhibitions which is from 2.4–19.5 (in no deviation case) across the data sets decreases to 1.7–6.3. Removing the region constraint slightly increases the network size (1.35 to 2.52 time). Increasing the deviation to 2 has few impact on the activation/inhibition distribution but greatly increases the total number of predicted regulations (up to 7.39 time larger networks, the increase is greater when the cell types are more different than when they are more closely related). We also note that increasing the network by relaxing the constraints by 1 includes more genes in the inferred relations, with no further increase when the deviation is set to 2 (S7 Fig). This suggests that a deviation of 1 with the region constraint is the best compromise between network size and expressed gene inclusion in the networks.

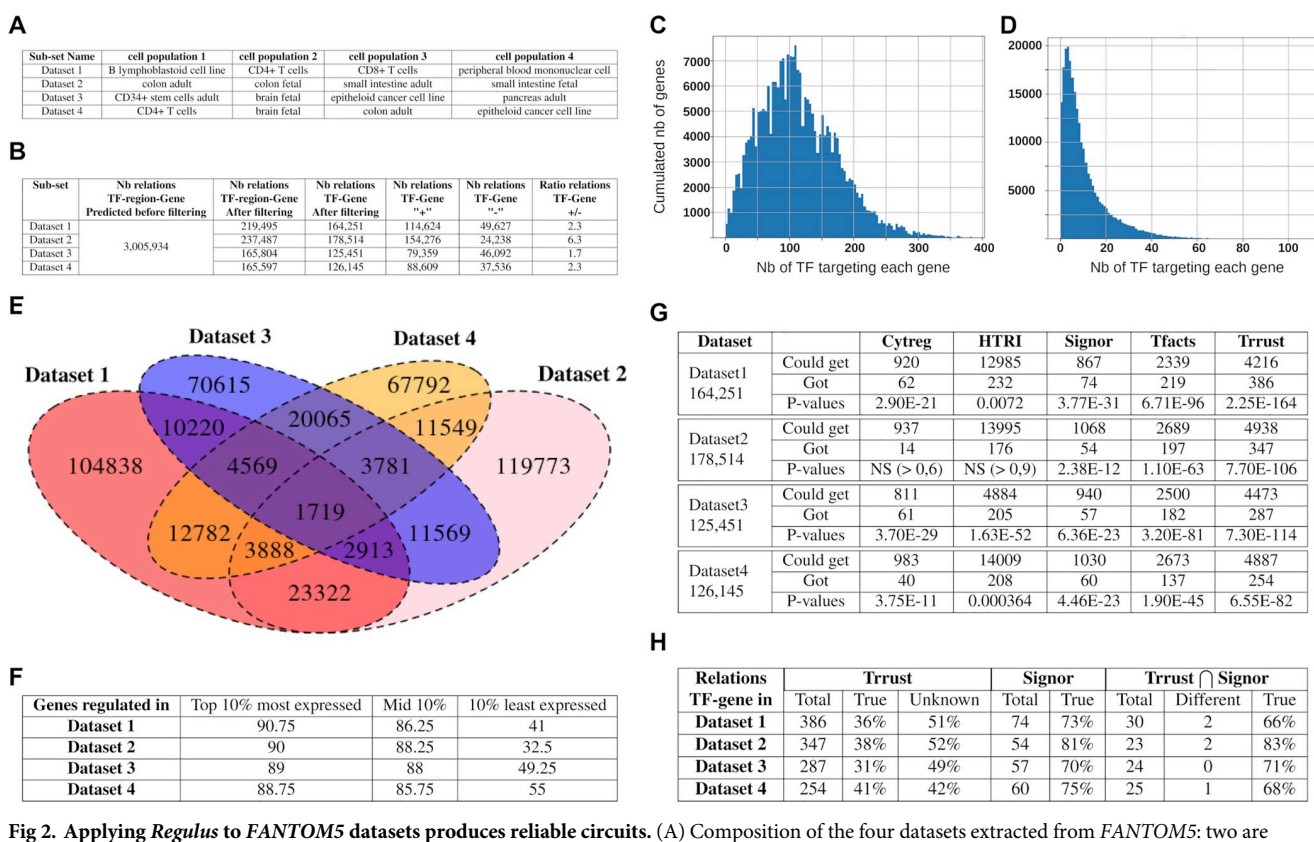

**Fig 2. Applying *Regulus* to *FANTOM5* datasets produces reliable circuits.** (A) Composition of the four datasets extracted from *FANTOM5*: two are composed of biologically-similar cell populations and two are composed of dissimilar cell populations. (B) Number of relations inferred by *Regulus* for each of the datasets depicted in A. (C-D) Distribution of the number of TFs regulating an expressed gene in *Regulus* circuits before (C) and after (D) the filtering using likelihood constraints. (E) Overlap of the relations found in the regulatory circuits corresponding to the four datasets of A. (F) Percentage of genes from *Roadmap Epigenomics* RNA-seq datasets related to the cell populations found in *Regulus* inferred circuits. Genes are separated in three categories according to expression levels: the top 10%, the middle 10% and the bottom 10%. (G) For each dataset, number of relations which were present in each of the five database ("reachable"), number of these relations which were inferred by *Regulus* ("found") and p-values of enrichment for existing relations in *Regulus* inferred circuits, as assessed by binomial test. (H) Relations found in Trrust and Signor and coherence of signs. *True*: percentage of relations found with the same sign as in the database, *Unknown*: relations non signed or signed + and—in the databases. For the intersection of Trrust and Signor: *Different*: number of relations signed differently in the two databases, *True*: percentage of relations having the same sign after inference by *Regulus* as in both databases.

**Inferred relations are validated by public data.** **Relations existence**. To validate the relations inferred using *FANTOM5* data, several databases are queried: Trrust [22], Signor [23], CytReg [24], HTRIdb [25] and TFacts [26] (Databases were queried on the 08[th] Oct. 2021). These databases contain manually curated TF-gene relations, found in cell types which may not be related to *FANTOM5* datasets and only represent a fraction of all possible relations. Therefore their content is one or two orders of magnitude smaller than the size of the circuits inferred by *Regulus* and the number of common—or reachable—relations is even lower (Fig 2G). Altogether, our results show that *Regulus* inferred circuits obtained before the constraints filtering step are highly enriched in relations described in several databases (p-values ranging from 1.05e-59 to 2.25e-164). Applying the likelihood constraints mostly preserves the enrichment in relations found in these databases (p-values ranging from 0.0072 to 2.75e-288, Fig 2G).

**Relations signs**. Trrust and Signor both contain signed relations, but some can be unsigned or contradictory signed (within the same resource or between both), as activation or inhibition depends on the biological context. As shown in Fig 2H, the signs of the relations mentioned in

both Trrust and Signor are the same in most cases and for 72% of these relations, this sign is correctly predicted by *Regulus*. For the relations generated by *Regulus* and found at least in one of Signor or Trrust, the predicted sign is consistent with the databases in two third of the cases.

We note that the context can actually determine the sign of the relations. When comparing datasets 1 and 4, we show that 22,958 relations are common between both datasets, among which 4,126 have an opposite sign. For example, AHR positively regulates *CYP1A1* in dataset 1 whereas AHR inhibits *CYP1A1* expression in dataset 4. This could indicate different metabolic functions in different cell types. Another example is the opposite regulation of *CD274* (encoding PD-L1) by BACH2 in datasets 1 (lymphoid cells) and 2 (intestinal tissues). This suggests that *CD274* could be differentially regulated by BACH2 in different cell types, or that immune cells of the gut have specific regulatory circuits.

**Impact of the deviation and region likelihood constraints**. For the number of relations found in the different databases, we observe a linear relationship with the total number of relations (S8 Fig). As lowering the likelihood constraints increases the total number of relations, it correlates with an increase in the relations which can be found in databases.

For the signed relations, we note that relaxing the biological likelihood constraints slightly decreases the quality of sign predictions when considering the TRRUST and Signor databases (S9 Fig). This suggests that the quality of predicted signs is better when likelihood constraints are more stringent.

## Comparison between *Regulus* and *Regulatory Circuits*

After validating our pipeline, we compare it to *Regulatory Circuits*, the closest method (and the only one which could be implemented) of circuit inference in terms of dataset size and nature of the genomic features, using the same *FANTOM5* datasets as in the previous subsection.

**Workflows comparison.** Fig 3 presents the main steps of the *Regulatory Circuits* pipeline (Fig 3A) and compares them to those of *Regulus* (Fig 3B). Both take as input the regions and genes activities and localizations, as well as TF binding sites coordinates. Both also share similar pre-processing steps like computing the distance between regions and genes or finding the TF binding sites occurrences in the regulatory regions. The main differences are: (1) *Regulus* uses the activity of the TF (extracted from its coding gene activity) which is not taken into account by *Regulatory Circuits* workflow; (2) *Regulatory Circuits* uses a composite score in which each component must be strictly positive, and is taken as a maximum when several concordant relations exist. This approach brings a bias towards activation relations and favors highly expressed genes. On the contrary, *Regulus* checks the global biological constraints between the different activities of the relation entities over all cell populations to produce signed circuits; (3) *Regulatory Circuits* gives cell type-specific circuits whereas *Regulus* outputs circuits by patterns, adding dynamics to the network, although cell type specific relations can be found by looking at the corresponding patterns.

**Circuits and outputs comparison. Relations found in reference databases**. The same TF-gene relations databases as above are used to evaluate the results of *Regulatory Circuits*. Overall, *Regulatory Circuits* produces similar enrichment of relations found in databases as *Regulus*, with significant enrichment obtained in 19 out of 20 cases (p-values ranging from 4.6e-3 to 1.7e-290, see S2 Table).

**Circuits topology**. For the four *FANTOM5* datasets used, *Regulatory Circuits* number of potential TF-genes relations (2,060,960) is calculated based on the supplementary data files describing entities (TF, promoter, enhancer, transcript) relations, while ignoring the scores [10]. All the TF-gene relations found by *Regulus* are included in the potential *Regulatory Circuits* relations, meaning that our method does not create irrelevant relations.

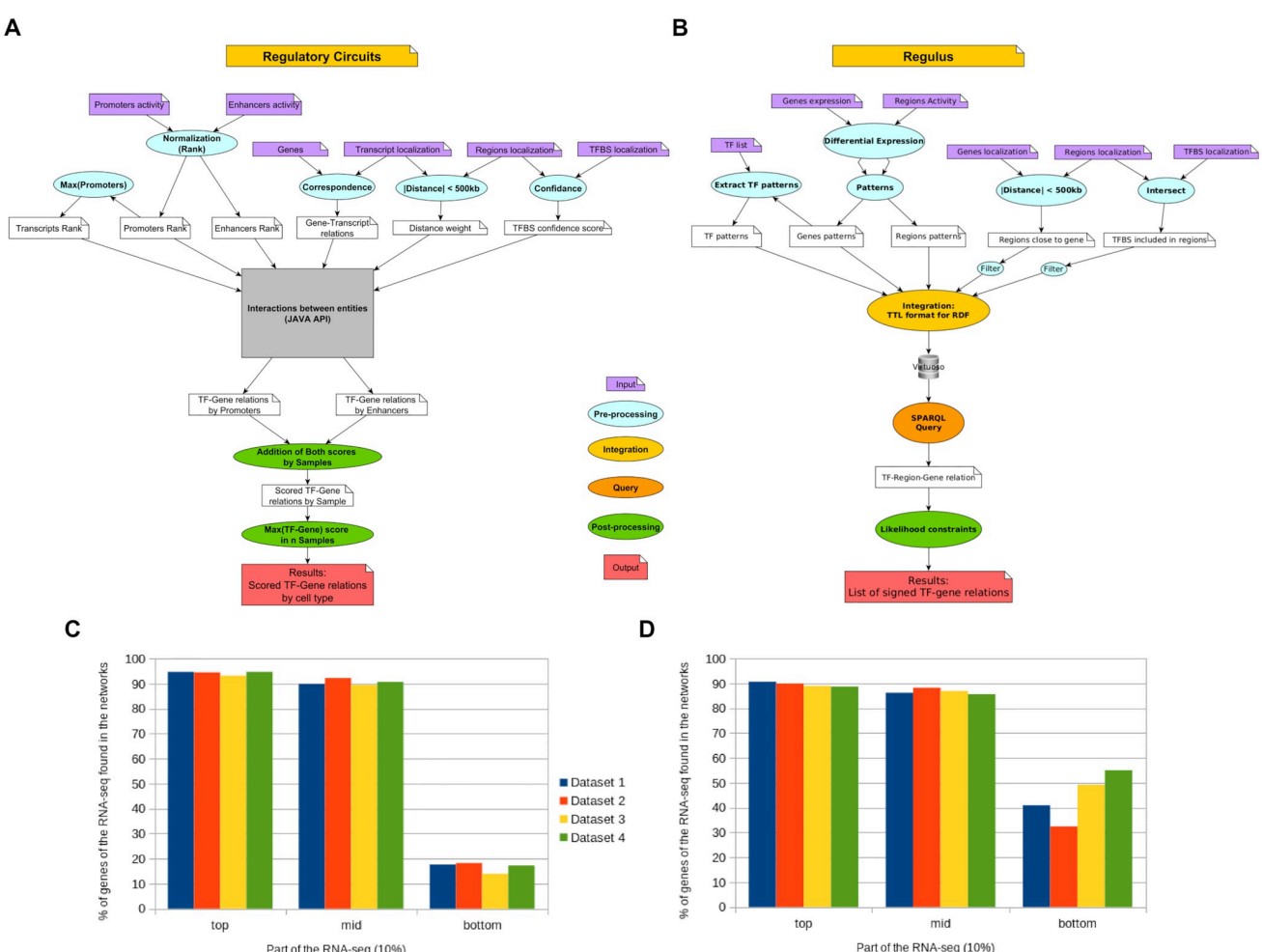

**Fig 3. Comparison of the *Regulatory Circuits* (A) and *Regulus* (B) workflows.** The pre-processing (in blue) steps are similar: normalization of the expressions, limitation of the distance between region and genes and inclusion of the TFBS in the regions. The main differences are: *Regulatory Circuits* uses a rank normalization of entities activities while *Regulus* uses a clusterization by patterns. While the construction of the regulatory circuits use different technologies (gray, yellow and orange) they follow similar principles: finding the biological relations between the entities and retrieving the weight (A) or patterns (B) along the relations. The main differences are in the post-processing (green): *Regulatory Circuits* gives a score for each relation in a sample, filters out the null scores and computes circuits at the cell type level, while *Regulus* computes a regulatory circuit for each expression pattern, filters it with biological constraints and signs every interaction. (C-D) Comparison of gene inclusion into regulatory relations depending on their expression level when *Regulatory Circuits* (C) or *Regulus* (D) is used to infer the relations.

However, *Regulatory Circuits* provides very large circuits, composed of 407,056 to 1,796,098 unsigned relations for a single cell population; whereas *Regulus* returns fewer relations (less than 180,000 for sets of four different cell populations), adding an significant filtering power, and is able to qualify them as activations or inhibitions. Interestingly, *Regulatory Circuits* relations show the same "number of TFs per gene" distribution than *Regulus* intermediary output before filtering by likelihood constraints (S10 Fig top).

**Inclusion of genes based on their expression level**. When using a similar validation of output genes inclusion using Roadmap Epigenomics RNA-seq datasets, *Regulatory Circuits* is able to retrieve highly or moderately expressed genes in its circuits. However, lowly expressed genes are poorly incorporated, in agreement with the methodology favoring inductive relations and high activities (see above). On the contrary, *Regulus* manages to retrieve similar

percentage of highly and moderately expressed genes but higher percentage of lowly expressed ones (Fig 2F and S10 Fig bottom).

## Application to B-cells identifies known and potential new key regulators

After showing that *Regulus* performed as well as or better than *Regulatory Circuits* when used on the same *FANTOM5* datasets composed of few samples and including closely related cell types, it is applied to B cell differentiation datasets to explore the use case presented in *Introduction*.

**Biological context.**    After stimulation by a pathogen, naive B cells (NBC) differentiate into either memory B cells (MBC) or plasmablasts (PB). MBCs store some information about pathogen encounters and are able to differentiate faster and more efficiently if the same pathogen is present again. PB are effector cells and produce antibodies to inactivate and eliminate the foreign pathogens. Regulatory circuits are required to explain the different abilities of NBC and MBC to differentiate into effector cells. For this study, four distinct populations are used: NBC, IgM$^+$ MBC (MBC IgM), IgG$^+$ MBC (MBC IgG) and PB. Gene expression data (RNA-seq, 26,734 genes) and chromatin accessibility data (ATAC-seq, 58,848 regions, used to determine regulatory regions activities) were generated. In this specific case the four populations are sequential: NBC is the first population and PB is the last, but MBC can be either a transitional state or a final one. Entities patterns will therefore have four elements, representing the relative entity activity in one B cell susbset in the following order: NBC, MBC IgM, MBC IgG and PB. Three main TFs are highlighted in the bibliography at different steps of the differentiation: an inhibitor, BACH2, and two activators, IRF4 and PRDM1 [8], as shown in Fig 4A. We thus expect that *Regulus* infers these TFs as main regulators in this system and in particular, that BACH2 is identified as an inhibitor of gene expression patterns increasing specifically in PB (such as 1114 or 1124, see Methods), but as an activator of decreasing gene expression patterns (4441, 4331 or 4431 for instance). PRDM1 and IRF4 are expected to have the opposite actions.

**Patterns are indicative of expression dynamics.**    After differential expression analysis, 14,921 genes have a null or very low expression level in all samples and 3,591 genes are not differentially expressed in any comparison. These genes are respectively attributed the 0000 and 5555 expression patterns according to the convention described in the Method subsection *Gene expression and region accessibility patterns*. The remaining 8,222 genes are distributed over 107 patterns—among the 192 possible. As shown in Fig 4B and S3 Table, these patterns contain from 1 to 1,418 genes and 18 patterns are composed of more than 100 genes. It can be noticed that gene expression is not randomly distributed and that patterns indicate the main dynamics at work in our system: the most numerous patterns by far are 4441 and 1114. They show that many genes are down-regulated when either NBC or MBC are driven towards differentiation into PB—the B cell identity genes—whereas another set of genes, necessary to mediate the PB differentiation and identity transition is induced [8]. Interestingly, the third most numerous pattern is 1124—containing the well known regulator PRDM1, which denotes the same dynamic as 1114, but with a small increase in IgG+ MBC, underlining their more advanced state of differentiation.

Among the genes, TFs are treated as a specific class of entities since they are the potential regulators to be identified by *Regulus*. In our dataset, 611 TFs are retrieved from the 643 for which binding site information is available in our dataset. 336 TFs are unexpressed and hence will not impact the circuits, leaving a set of 275 potential regulators in our system. TF expression patterns distribution is heavily unbalanced, since they are present in only 58 patterns. Moreover, 56 are constant genes, 63 have the 4441 pattern and 15 have the 3441 pattern,

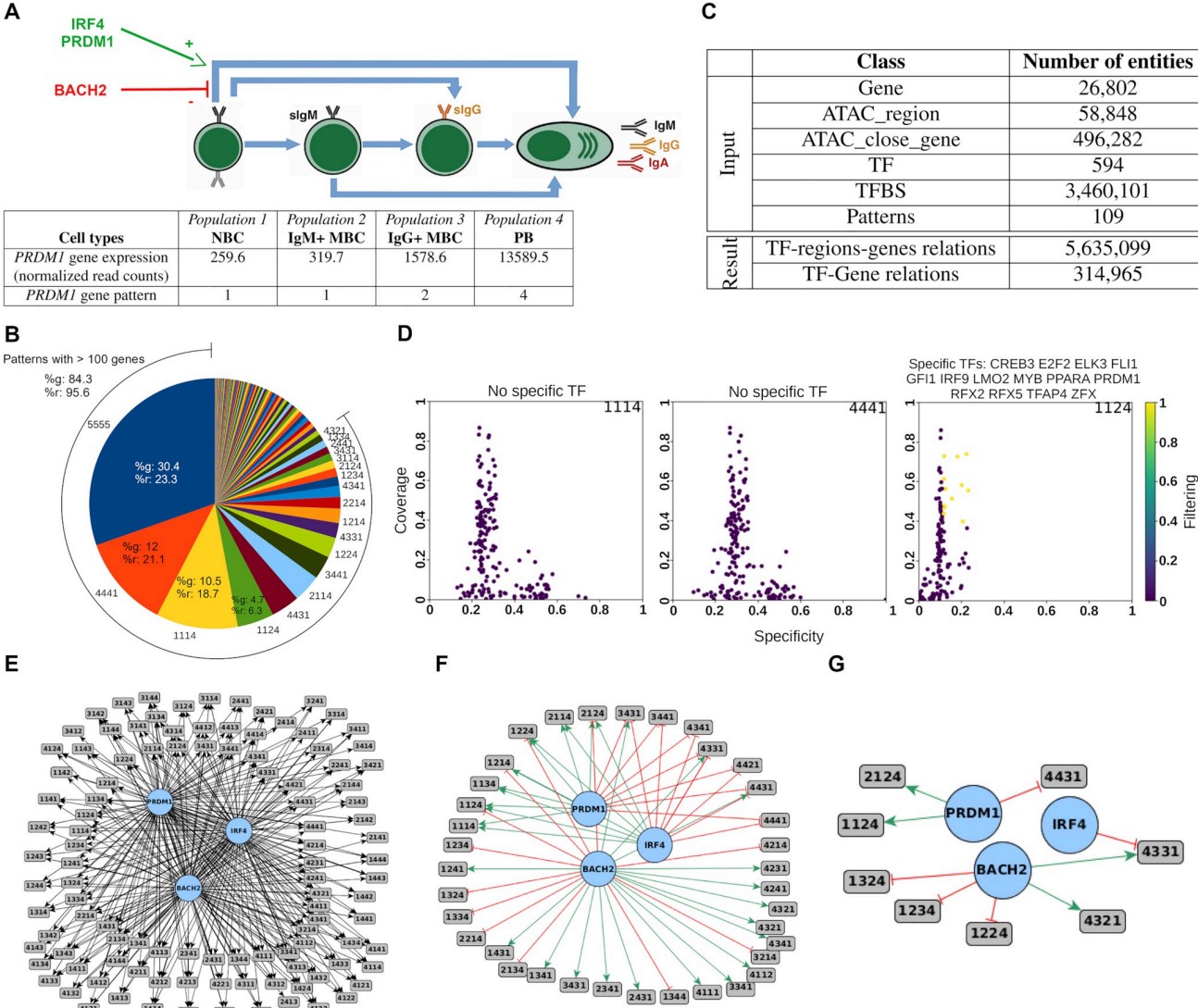

**Fig 4. The B cell regulatory circuits computed with *Regulus*.** (A) Biological interaction between the cell types and main known regulators of the B cell differentiation process. Black arrows show the possible transitions between the different cell types. Cell types are ordered from the less differentiated (NBC) to the most differentiated (PB). BACH2 is a negative regulator of this process, whereas PRDM1 and IRF4 are described as positive regulators. Below the cells, an example of gene expression and its associated pattern is shown. The first pattern element corresponds to NBC, the second to IgM+ MBC, the third to IgG+ MBC and the last to PB. NBC: naive B cells, MBC: memory B cells, PB: plasmablasts. (B) Table showing the main inputs and outputs of *Regulus* for the B cell differentiation regulatory circuits. (C) Pie chart showing the number of genes per patterns of gene expression. The arc represents the 18 patterns with more than 100 genes, %g: percentage of total genes included in this pattern, %r: percentage of total relations targeting a gene included in this pattern. (D) Distribution of the coverage and specificity for all the TFs targeting 1114 (left), 4441 (middle) and 1124 (right): yellow dots indicate the TFs passing the thresholds. While being the largest patterns 1114 and 4441 do not have any key regulator with our filtering method. (E-G) Graphs of interaction of the main known regulators and their targeted patterns before (E) and after (F) filtering the total relations with the likelihhod constraints and (G) after filtering with coverage and specificity: relations are consistent with the known roles of BACH2, IRF4 and PRDM1 during the PB differentiation.

whereas only 11 and 13 TFs are included in the 1114 and 1124 pattern, respectively. This suggests that most of the regulation supporting the NBC / MBC differentiation towards PB is due to TFs which expression is extinct in plasmablasts and is mainly a release of inhibitions allowing the PB fate driving regulators and genes to be expressed.

Finally, regulatory regions are divided over 110 patterns with a distribution which is highly correlated to the gene expression patterns ($r^2 = 0.89$ including all patterns, $r^2 = 0.68$ when the

constant patterns are removed). As for the genes, the 4441, 3441, 1114 and 1124 patterns are the most numerous, underlining that according to the biological rules of gene regulation, the above-mentioned gene expression inhibitions and inductions are sustained by concordant changes in regulatory regions accessibility.

In summary, pre-processing the data into patterns provides some filtering power, allows the user to concentrate on patterns relevant to the biological context and gives a first glance of the dynamics at work in B cell differentiation.

**Potential relations need to be filtered by biological constraints.** Pre-processed data are integrated to create the data structure and the relation graph and then queried. *Regulus* inputs and outputs are shown in Fig 4C. The pipeline generates 5,635,099 TF-regions-genes relations. Filtration by the biological likelihood constraints (with a deviation set to 1 and taking the region effect on gene expression into account, according to the previous section results) outputs 612,633 signed TF-gene relations. These relations are merged to 314,965 unique relations by regrouping identical TF-gene relations occurring through different regions. Of those relations 173,717 are signed as activation (+) and 141,248 as inhibition (-). As described in Fig 4B and S3 Table, the most frequent gene expression patterns (except the constant one) have a tendency to aggregate a larger proportion of relations compared to the percentage of total genes they include. As shown in Fig 4C and by comparing Fig 4E and 4F, the likelihood constraints refine the resulting circuits to the potentially most relevant relations.

**Key regulators can be identified by coverage and specificity filters.** As a further refinement to identify the most biologically relevant TF-gene (or TF-pattern) relations, the stand-alone tool *ClassFactorY* is provided, allowing to apply a coverage and specificity filter and to provide an annotation-based score (see Methods). After setting the specificity and coverage thresholds to the last quartile, no specific TF is highlighted for 25 patterns, a single potential regulator is found for 16 patterns and 10 or more regulators of interest are identified for 35 patterns. Surprisingly, for the two non-constant most populated patterns (1114 and 4441), no TF could be selected by the coverage / specificity filter (Fig 4D). Out of the 238 TFs found in the resulting relations, 116 do not pass the thresholds for any pattern, 23 regulate only 1 pattern and 19 are associated with 10 patterns or more. The combination of both parameters also allows us to filter out some highly ubiquitous TFs, such as SP4.

Among the 121 TFs passing above the threshold, many have already been described as implicated in B cell differentiation, including the main known regulators PRDM1, IRF4 and BACH2 [8]. At the pattern level (Fig 4G), IRF4 is found as an inhibitor of 4331, PRDM1 is an activator of 1124 and 2124 (strong expression specifically in PB) and an inhibitor of 4431 (low expression in PB). BACH2 is identified as an activator of 4321 and 4331 (decreasing expression during differentiation) and an inhibitor of 1224, 1234 and 1324 (increasing expression during differentiation). *Regulus* results are therefore in agreement with the literature and with our expectations. Complete metrics about TF-gene relations based on the number of genes in each pattern and describing the number of relations, of potential regulators before and after the coverage / specificity filter and some biological interpretation are available in S3 Table. The refining power of both the likelihood constraints and the coverage / specificity filter on the identification of regulators for genes sharing a similar expression dynamic (i.e. pattern) is appreciated by comparing the resulting circuits in Fig 4E–4G.

**Regulators annotation as a decision helping step.** *ClassFactorY* also includes an annotation based score (see Methods subsection *Finding key regulators with ClassFactorY*), based on GO terms, Pubmed citations and MesSH terms allowing the validation of already known TFs, relevant in the biological context, and providing a list of potential TFs of interest for further biological investigation. Out of the 121 TFs identified in the previous step, 64 are annotated by

the GO term "cellular developmental process", 27 by "immune system process", 13 by "lymphocyte activation", 3 (IRF8, LEF1, YY1) by "B cell activation", 2 (IRF8, YY1) by "B cell differentiation" and none by "plasma cell differentiation". The number of citation in Pubmed for the found TFs ranges from 164,841 (MAX) to 2 (ZNF75A).

Citations involving MeSH terms relative to the specific context of the B cell differentiation and each of the TFs are counted: 93 TFs are cited, among which IRF4, STAT3, MYC, PRDM1 & PAX5 being the five with the most citations (208 to 386). A second query is done with a more global context relative to B cells in general: 104 TFs are annotated; PRDM1, PAX5, STAT3, MYC & MAX are the five ones with the most citations (402 to 2573). A null score is attributed to six TFs (KLF16, FOXJ3, TFAP4, TGIF1, ZNF219 and ZNF75A), meaning that although they have been selected by the coverage / specificity filter, their potential role in B cell has not been studied yet and may be worth investigating.

## Discussion

In this article we present a new design for regulatory circuit inference when few samples describing different cell states are available. *Regulus* addresses the methodological issues presented in Introduction: (1) the under-exploitation of the regulatory context (limited or no use of TF binding or region accessibility data), (2) data reduction and structure, (3) the lack of functionality qualifier for interactions (activation of inhibition) and (4) pipeline availability for reuse and reproducibility. To solve theses issues (1) the linked and inter-dependent data about gene expression, TF availability and region accessibility are used, (2) genes or regions of similar activity trend are clustered under the same pattern, (3) Semantic Web technologies support data integration, browsing and querying. A global network based on all the samples is computed and the biological likelihood of the relations is checked, allowing to sign the relations, introducing a significant proportion of inhibitions in the final network, (4) a standalone automated version of *Regulus* is provided to facilitate its reuse (https://gitlab.com/teamDyliss/regulus).

Conceptually, most regulatory circuits are inferred by computing correlations between genes and TFs, by adding edges if the correlation is higher than some threshold and, in the most refined cases, by keeping the edges for which the TF binding site is accessible. However, this approach can only be used for large datasets as correlations are highly sensitive to outliers and therefore not very reliable when applied to few samples [12]. Our main hypothesis is that linked information contained in small numbers of samples can be exploited by discretizing activity and gene expression levels into uniform bins. Although it may be necessary to modulate the number of bins to fit situations with a different number of compared cell types, it should be kept in mind that increasing the number of bins will exponentially increase the number of patterns. This could make the activity dynamics difficult to follow and introduce complexities when applying the biological likelihood constraints to these patterns. On the other hand, lowering the number of bins will reduce the ability to apply the likelihood constraints, as it will result in less filtering power by reducing the number of pattern levels combinations. This simple discretization scheme allows to handle the variability in entities activity since each feature is considered individually on the different available samples. Based on the discretization procedure, *Regulus* general approach of inferring signed relationships is qualitative: likelihood constraints based on the effect of regulation and site accessibility on gene expression. The inferred sign is naturally deduced from the constraints application and is only reported when the relationship is consistent in all considered cell populations.

As a perspective work, inferring the likelihood constraints in a data-driven manner could be investigated, as well as defining the best binning procedure depending on the number of

compared cell types. This would be particularly interesting to devise rules that could applied to cases with a larger number of samples, when the number and the complexity of patterns are increased. In its current state, this is the main restriction for applying *Regulus* to studies with many samples. The fully qualitative approach and the cautious criteria used to infer a sign is a way to ensure the robustness of our approach to noise. In the case when sample number is low, the strategy used by *Regulus* therefore transforms the regulatory circuit inference statistical inference problem into a reasoning issue which can be handled with formal methods.

Describing the expression or activities as patterns also allows the user to concentrate on patterns which have a biological meaning in the given experimental setting. Cell types are not individualized by these patterns, but biologists are often more interested in dynamic changes between cellular states than by a complete record of regulators active in a fixed cell population. Even if they are interested in such characterization, it is possible to look at regulators for the pattern where expression is at its highest only in the cell population of interest. A limit of this approach might be the over-sampling of the patterns as some were scarcely populated in the B cells data. Those patterns can be merged with patterns of similar trend or removed from the analysis, since they may bias the coverage and specificity filters. Indeed small patterns are easily covered and result in high specificity percentages. Another solution would have been to use co-expression analysis for example using the WGCNA R [27] package. After testing this approach on our limited number of B cells datasets, it gave poor results, clustering together genes which have described opposite functions, such as *BACH2* and *PRDM1*.

*Regulus* uses both RNA-seq and one measurement of regulatory regions activity. This can be obtained either from ATAC-seq, as shown in this study or from other techniques such as DNAse or MNase sensibility, active histone marks ChIP-seq or FAIRE-seq. ATAC-seq is now widely used, requires only 50,000 cells and is accessible to most molecular biology laboratories. ATAC-seq data can also be found in large scale genomic databases, such as the ENCODE project which contains ATAC-seq data for 157 cell types. Therefore, *Regulus* allows to reuse these data when laboratories are not able to perform ATAC-seq on their samples.

Our work shows the added value of integrating heterogeneous linked and inter-dependent data from biological experiments when inferring regulatory circuits. Even on closely-related cell types, such as the B cells subsets, subtle differences can be identified owing to the use of patterns and the biological likelihood constraints filtering. Semantic Web Technologies allow for an easy identification of relations. Once the data structure is obtained, it can be queried to answer any specific question. As it is based on unique identifiers and self-structuring, it also reduces the risks of introducing false relations. This confirms that Semantic Web technologies, which were instrumental in the expansion of the Linked Open Data initiative [28], and in particular in life science data integration [18, 29], are also a suitable framework for supporting gene-regulatory circuit inference in the context of personalized medicine.

The *ClassFactorY* tool is provided to identify the key regulators by introducing a filter of coverage and specificity, and by adding biological context annotations. From the 237 potential TFs involved in B cell circuits, coverage and specificity reduce this number to 121, which is still too much to perform experimental validation. Annotations can thereafter be used on these "short-listed" TFs to (1) validate known regulators and (2) identify potential new regulators which have not been described in the context of interest. A perspective is to reduce the number of candidate regulators by using constraint programming to determine the smallest group of TFs able to regulate the biggest part of a gene pattern. A further refinement of this tool would be to introduce logical reasoning to delve into the dynamics or combined effects of TFs. Cooperative, competitive or exclusive binding of several TF at the same binding site can indeed occur and have different outcomes on gene expression, depending on the sequence of events and the involved TF. This is not inferred by *Regulus* in its current state.

Finally, the combination of *Regulus* and *ClassFactorY* is able to retrieve the main regulators of the B cell differentiation process, such as PRDM1, IRF4, BACH2 and PAX5 and to pinpoint them with a high annotation score. In addition, six new potential TFs impacting this process, identified through high coverage and specificity coupled to a null annotation score, will need to be further investigated, showing the power and interest of our tool.

## Methods

The numerical data used in Figures are included in file S1 Data.

### Main characteristics of *Regulus*

As inputs, *Regulus* requires: (1) a list of genes with their expression in a selected number of cell types, (2) a list of selected regulatory regions with their activity as two text files, (3) genomic locations of genes, regions and TF binding sites as bed files. The latter can be provided by the user (ChIP-seq data for a specific TF for example), but the genome-wide TF binding sites coordinates from *Regulatory Circuits* [10], containing curated binding sites for 643 TFs, are provided. For the purpose of our study, all genomic coordinates are given according to the *hg19* human reference genome. It was indeed used in the *Regulatory Circuit* project, with which we wanted to compare *Regulus*. As a convenience for other purposes, we also provide a list of TF binding sites lifted over the *hg38* reference genome. Users are not dependent upon a specific reference genome, as long as they have genomic coordinates for genes, regions and TF binding sites of interest in the same species. *Regulus* outputs a list of candidate signed TF-genes relations with patterns indications for all entities, that can be explored to identify new regulators.

### Pre-processing

**Neighborhood relationship.**   Genomic coordinates are used to compute base pair distances between regions and genes with the *distance.py Python* script. The distance is calculated between the two closest extremities of the entities, regardless of their respective position. All distances are filtered at a max threshold of 500 kb and set to 0 for overlapping entities.

**Including TF binding sites in regulatory regions.**   *Regulatory Circuits* [10] data on TF localization across the genome are used, as they contain reliable and extensive information. The *Bedtools intersect* [30] tool is first used to identify all the TFs binding sites included into a set of regions. Then, binary relations between TFs and regions are produced by only keeping the information about a given TF binding in a definite region at least at one binding site.

**Gene expression and region accessibility patterns.**   Entities (genes, TFs or regions) with differential activities (expression or accessibility) are defined by the user as input and grouped into patterns, according to activity dynamics. This discretization is performed independently for each entity, as activity levels may greatly differ between entities [10]. A pattern is a *n*-tuple with *n* equal to the number of compared cell populations. First, for a given gene or region, for each population $i$, the mean per population based on samples (one population = several samples) activities (normalized read counts, reads densities...) is calculated and log-transformed. We denote by $g_i$ (respectively $r_i$) the gene expression (respectively region accessibility) log-transformed average value among replicates. The interval between the maximal and the minimal of these values is divided into four equivalent intervals, providing a scale from 1 (minimum) to 4 (maximum). The number of four bins was chosen based on the features activities distribution, which were mainly separated by 3 to 5 equal bins (S2 Fig). Each averaged expression value, and therefore each cell population, then gets an attribute from 1 to 4 corresponding to the interval where this value belongs. For each measurement $g_i$ (or $r_i$) in a series of *n* cell populations, the value attributed by this procedure is denoted *level*($g_i$) and calculated as

follows:

$$level(g_i) = \max\left(1, \left\lceil 4\frac{g_i - \min_{k \leq n}\{g_k\}}{\max_{k \leq n}\{g_k\} - \min_{k \leq n}\{g_k\}}\right\rceil\right) \in \{1, 2, 3, 4\}$$

As an example Fig 5 shows the pattern attribution of an entity activity over four cell populations, in the form of a quadruple of integers, where the entity is a gene and its activity measure is its averaged expression. Fig 5A shows the gene expression (in normalized read count per million) and the attributed pattern value for each cell population. Fig 5B illustrates the pattern generation: the highest activity level (log10 transformed) is set to 4, the lowest is set to 1 and the space between both is divided in four equal intervals, numbered from 1 to 4. Each cell population gets the interval number corresponding to its activity level as a pattern value, and these values are aggregated in a quadruple, in this case leading to the pattern 1124. This step is performed by two *Python* scripts, *gene2pattern.py* and *region2pattern.py*.

Finally, entities with no or low activity (as defined in differential expression analysis for genes, for example) in all samples are granted a profile with only zeroes as values and are

## A

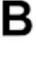

| Cell types | *Population 1* **NBC** | *Population 2* **IgM+ MBC** | *Population 3* **IgG+ MBC** | *Population 4* **PB** |
|---|---|---|---|---|
| Gene expression (normalized read counts) | 259.6 | 319.7 | 1578.6 | 13589.5 |
| Gene pattern | 1 | 1 | 2 | 4 |

## B

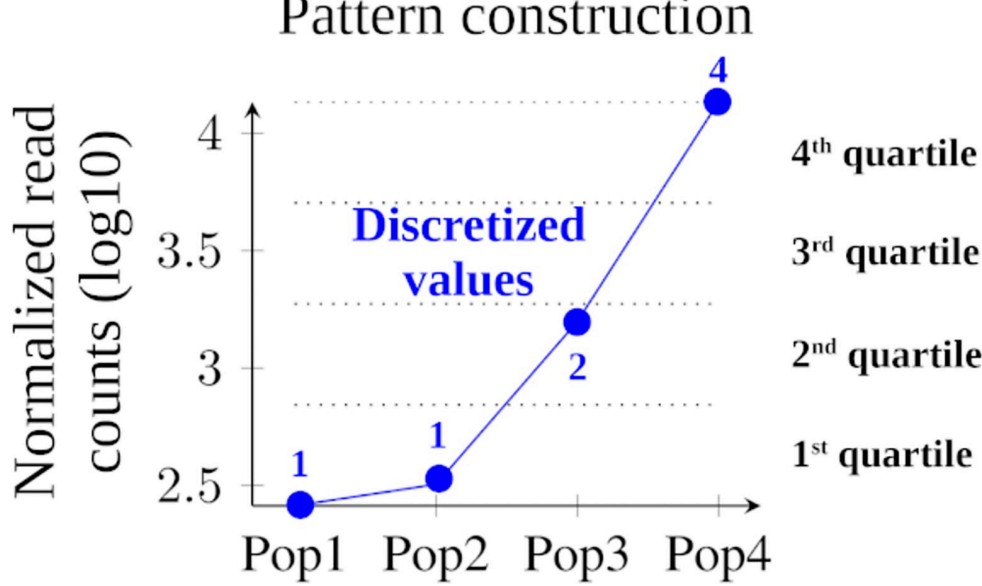

**Fig 5. Example of a gene expression pattern construction.** (A) This gene expression is characterized by two low expression values followed by an intermediate and a very high value. (B) This expression is modeled by the 1124 pattern after log transformation, binning of the difference between minimal and maximal values in four equal intervals and assigning the corresponding bin number to each expression as its discretized value.

removed from the implementation, since they do not bear any relevant biological information. A profile with *5* for all cell populations is attributed to entities with constant activity. The TF patterns are those of their respective coding genes.

## Data graph for integration and query

Pre-processed data are transformed from tabulated to TTL format by the *csv2ttl.py Python* script, allowing to introduce the inclusion relation between TFs and regions and the distance relation between regions and genes. Then, *Regulus* generates a structured RDF graph of data that can further be queried with SPARQL (S3 Fig).

The construction of the RDF dataset requires the introduction of unique identifiers and reified entities [31, 32] to describe some relations. *Identifiers*: for the regions, a unique identifier is designed after the type of region (i.e. ATAC_; and Region_ in the text) followed by the row number at which they appear in the region localization file—this ensures that a same number is never used twice for different entities. For genes and TFs, their HGNC Gene Symbols are kept as identifiers. *Reified entities*: as RDF does not allow relations to bear a score, relations of distance between genes and regions and of TF binding into regions are both inserted by using reified relations. This method generates new entities with dedicated identifiers and bearing either the distance value or the inclusion. *Additional information*: references to Uniprot and Ensembl are added for the genes to link our data with public databases. The genomic localization information for all entities is also kept, as a potential filtering parameter.

To find the potential regulators of a given set of genes, *Regulus* uses a SPARQL query (S4 Fig) on the data graph generated previously to extract all TF-region-gene triples and their related patterns. This step is embedded in the *sparql.py Python* script.

## Regulatory likelihood constraints

**Theoretical principles.**    To reduce the space of inferred relations, *Regulus* relies on biological likelihood constraints, such that the predicted relations have to be supported by observational *n-tuples* (e.g. patterns) compatible with the biological principles of regulation. Basically, for a positive (respectively negative) regulation, if the TF is present and its binding region is accessible, then the gene expression is high (respectively low). Then the regulation strength of the TF is modulated by the accessibility of the region: the more the region closes, the more the TF have to be expressed to output the same gene expression level. Starting from the situation where the region is fully accessible and therefore allows all possible regulations, we devised "*perfect*" regulatory relations, where the constraints are satisfied for each element of the TF expression, gene expression and region accessibility *n-tuple* patterns.

To this aim, we consider that observations support a positive regulation from a transcription factor *f* to a gene *g* through a region *r*, when the triple of *n*-tuples TF-expression $(f_i)_{i \leq n}$, region accessibility $(r_i)_{i \leq n}$ and gene expression $(g_i)_{i \leq n}$ measurements satisfies the relation (1) for each compared cell population:

$$\forall i \leq n, \qquad level(g_i) = \max(1\,,\, level(f_i) - (4 - level(r_i)))$$
$$\text{(positive regulation condition)} \tag{1}$$

This means that, for each cell type, the targeted gene-expression level (when compared to the expression of the same gene in the other cell populations) is consistent with the transcription factor expression level modulated by the accessibility of the region. As this constraint is satisfied for each considered cell population, there is a strong support for assuming that a single regulation occurs to explain the consistency between all the considered data.

However, gene-expression levels are obtained as a balance between the strength of the transcription factor regulatory effect and the accessibility level of the region. Therefore, a different distribution of the region accessibility densities may induce permutations of the gene expressions levels with respect to the distribution of transcription factor levels, compared to the "perfect" situation. We then added a relaxation factor to take this situation into account. We consider that a regulation from a transcription factor *f* to a gene *g* through a region *r* is supported by their transcription factor expression $(f_i)_{i \leq n}$, region accessibility $(r_i)_{i \leq n}$ and gene expressions $(g_i)_{i \leq n}$ if they satisfy the two following relations:

$$\exists (l_i)_{i \leq n}, \quad \text{s.t} \quad \forall i \leq n$$
$$\sum |level(g_i) - l_i| \leq \delta \text{ and } l_i = \max(1, \, level(f_i) - (4 - level(r_i))), \quad (2)$$
$$\text{(relaxed condition on expressed genes)}$$

$$level(g_i) \leq max(1 \, , \, level(f_i) - (4 - level(r_i)) + 2) \quad (3)$$
$$\text{(relaxed condition of regions accessibility)}$$

The first relation ensures that the distance between the observed triple and the perfect situation $l_i$ is smaller than the threshold $\delta$. The second relation ensures that the TF is sufficiently expressed and that the region is accessible enough for being able to draw a biologically-relevant conclusion.

Negative regulations are handled similarly with a symmetric constraint:

$$\forall i \leq n, \quad 5 - level(g_i) = \max(1 \, , \, level(f_i) - (4 - level(r_i)))$$
$$\text{(negative regulation condition)} \quad (4)$$

**Algorithmic implementation and testing.** To encode the above-mentioned principles, we represent the constraints as tables shown in S5 Fig. From any pair of TF expression and region accessibility patterns, it exists a unique gene expression pattern which completes them into a "perfect" (TF-region-gene patterns) triple for a positive regulation (respectively, negative regulation), as encoded by Eq 1 (respectively, Eq 4). This gene expression pattern can be computed with the tables shown in S5 Fig by selecting dark blue cells in the relevant table. Deviation from this situation either by a relaxed effect on genes expression (Eq 2) or on regions accessibility (Eq 3) is represented by the light blue cells and can be computed automatically. Removing any constraint linked to the loss of region accessibility (only using Eq 2 but not Eq 3) is represented by the grey cells.

To choose a relaxation factor ($\delta$) giving a balanced output between the size of the total network and the reliability (or likelihood) of the inferred relations, several values of $\delta$ are tested on the *FANTOM5* datasets and their effect on several qualitative metrics are described. To quantify the threshold, we arbitrarily set a value of 2 to the dark blue cells and to 1 to the light blue or grey cells of S5 Fig. The following thresholds are tested: no relaxation ($\delta = 0$, relations have to be compatible with the dark blue cells in all four cell populations), $\delta = 1$ (one light blue cell is accepted among the four cell population), $\delta = 1$ while relieving the constraints on the region (one light blue or grey cell is accepted, named *1_regOFF* in S6 to S9 Figs) and $\delta = 2$ (up to 2 light blue cells are tolerated). These settings are then translated into a filtering algorithm using the custom *tf_gene_all2consist.py Python* script.

### Finding key regulators with *ClassFactorY*

*Regulus* output tables can be merged to obtain unique TF-gene relations and refined by using the coverage / specificity and GO / MeSH annotation standalone external tool *ClassFactorY*.

**Coverage and specificity filters.**   The first filter applied by *ClassFactorY* is on coverage and specificity of TF for some gene patterns. *Definitions and computing*: coverage of a TF is calculated as the proportion of genes in a specific pattern which are targets of a given TF. The coverage itself does not provide enough information: the smaller patterns, sometimes composed of 1 or 2 genes, are easily fully covered by a TF. Specificity is based on the proportion of targets genes that are from a specific pattern, for a given TF. A TF has a high specificity for a pattern if out of all its targets a significant number comes from this pattern. As for the coverage, the specificity does not yield enough information by itself: despite having a large number of its targets in a pattern, a TF may have little influence on it if the pattern is large. Both the coverage and the specificity are calculated as percentages.

*Combination of coverage and specificity*: for a given pattern, a TF of interest is a TF which specificity and coverage are both superior to a threshold chosen by the user: mean + one standard deviation, quantiles or specific percentages. Then *ClassFactorY* outputs a list of selected TFs together with the gene patterns they potentially regulate.

**GO and MeSH annotation.**   To validate the TFs inferred by *Regulus*, the standalone module *ClassFactorY* allows for automatic queries of the GO, Uniprot and PubMed databases. A list of GO annotations provided by the user is used to verify if candidates TFs are annotated by these terms. MeSH terms are retrieved from a user-defined list of PubMed references about the biological context and the module counts the number of citations associating the TFs to one or several of the terms. An annotation-based score is then calculated and provided as a help-decision tool for end users.

### Datasets used for validation and testing

**Roadmap Epigenomics RNA-seq datasets.**   Gene inclusion in our circuits is validated with *Roadmap Epigenomics* RNA-seq datasets used in [10]. RNA-seq datasets corresponding to the *FANTOM5* cell populations used for circuit inference are separated in three gene sets corresponding to the 10% most expressed ones, 10% least expressed ones and the 10% in the center of the expression distribution. For each category, the percentage of genes which are included in the inferred circuits is reported.

**B-cells datasets.**   *Regulus* prediction abilities was investigated with datasets of differentiating B cells, comprising 3 replicates of RNA-seq for naive B cells (NBC), IgM secreting or IgG secreting memory B cells (IgM+ or IgG+ MBC) and plasmablasts (PB). Data are aligned on the hg19 human reference genome and gene expressions for 26,734 genes are calculated with *featureCounts*. Raw counts files are then used for differential expression analysis with *DESeq2* [33] in *R-4.0.0*. Low expressed genes are filtered out by the *R* package *HTSFilter*. Remaining genes are used for differential expression analysis by comparing each population again all the others. Variable genes are determined as those with an adjusted p-value $< 0.05$ and an absolute value of $\log2(\text{fold\_change}) > 1$ in at least one comparison. ATAC-seq data is obtained on the same cell types (n = 1), aligned on hg19 and accessible regions are called with MACS2 with a q-value cutoff of 0.001. An union of 58,848 regions called in at least one sample is computed with *bedtools merge* [30] by taking the widest boundaries for overlapping regions. For each sample, reads overlapping each region of this union are counted for with *bedtools intersect* [30] and normalized by sequencing depth and region size to compute read densities. This dataset, experimental procedures and detailed analysis settings are available on GEO (GSE136988 and GSE190458).

## Supporting information

**S1 Fig. Toy-example of Regulus pipeline.** Top panel: the pool of entities is composed of 2 TFs, 2 regions and 2 genes. All of the biological entities activities or expression are first discretized (see Methods subsection *Gene expression and region accessibility patterns*). Middle panel: the interactions between the entities are then computed thanks to their inter-dependency and relations, using a Semantic Web framework. Region2 includes no binding site for TF2 therefore this interaction is discarded and Region1 is more than 500kb away from G2 and does not pass the distance threshold. Then, TF-region and region-gene relations are combined to obtain TF-region-gene relations resulting in a putative unsigned network. Bottom panel: relations are then filtered using the likelihood constraints. For each pair of TF expression and region accessibility patterns, the gene expression pattern is compared to the unique "perfect" gene expression pattern compatible either with an activation or an inhibition (see Methods subsection *Regulatory likelihood constraints*). To include some flexibility in the regulatory relations caused by modulation in TF expression or region accessibility, a relaxation parameter ($\delta$) is introduced. Here the maximum allowed deviation ($\delta$) from a perfect TF expression, gene expression and region accessibility patterns triple is 1. This step attributes a sign to the relation as an activation (+) or an inhibition (-). The result is a filtered and signed network (bottom right box).
(PDF)

**S2 Fig. Number of bins for computing the activity patterns.** (a-b) For each variable and significant (over the minimal expression / activity threshold) gene expression or regulatory region activity of our B cell dataset, the number of equal bins necessary to separate all values between the minimum and the maximum was computed. The absolute frequency of this bins number is reported for genes (a) and regions (b). For most genes and regions, their activity over the samples is described by using three to five bins, with a peak at four bins. Relative to Figs 1 and 5 and Methods subsection *Gene expression and region accessibility patterns*.
(PDF)

**S3 Fig. RDF data structure model after integration of all pre-processed input data.** Example of the data structure obtained after applying Semantic Web technologies integration to B cells genomic datasets. Relative to Fig 1, Results subsection *The Regulus tool* and Methods subsection *Data graph for integration and query*.
(PDF)

**S4 Fig. SPARQL query used to retrieve all relations between TF, Region and Gene entities and their associated activity patterns.** Relative to Fig 1, Results subsection *The Regulus tool* and Methods subsection *Data graph for integration and query*.
(PDF)

**S5 Fig. Biological likelihood constraints threshold tables** divided by relation qualification as activation (left column) or inhibition (right column) and by region pattern value from 4 (highest accessibility value) to 1 (bottom, less accessible). Without deviation ($\delta = 0$), a relation is accepted only if for all cell populations, the TF expression, gene expression and region accessibility patterns triple satisfy Eq 1 in Methods subsection *Regulatory likelihood constraints*, corresponding to the dark blue cells. With a deviation of one ($\delta = 1$), relations are accepted if at most one cell population satisfy Eqs 2 and 3 (light blue cells), and all other cell populations satisfy Eq 1. A deviation of 1 and the relaxed constrain on the region ($\delta = 1\_reg-$ OFF) allows for only one cell population satisfying Eq 2 but not Eq 3 (light blue or gray cells) and the rest satisfying Eq 1. A deviation of two ($\delta = 2$) allows for either one cell population

satisfying none of the Equations above (white cells) or two cell populations satisfying both
Eqs 2 and 3 (light blue cells). Relative to Fig 1 and Methods subsection *Regulatory likelihood constraints*.
(PDF)

**S6 Fig. Construction of regulatory networks while varying the likelihood constraints deviation.** Constitution of the *Regulus* networks while shifting the deviation ($\delta$) level: 0 is the more stringent and 2 the more lenient one, deviation 1 and relaxed region constraints ($\delta$ = 1_reg-OFF) represent an offset of 1 all over the diagonal in the biological likelihood constraints table (see S5 Fig). The choice of a deviation of two lead to regulatory networks 2.4 to 7.4 times bigger than the most stringent one. We can also note that more relaxed constraints favor inhibition ("-") relations. Relative to Results subsection *Application to FANTOM5 data*.
(PDF)

**S7 Fig. Effect of likelihood constraints deviation on the recovery of known expressed gene.** Percentage of genes from *Roadmap Epigenomics* RNA-seq datasets related to the cell populations found in circuits inferred by *Regulus* using four different constraints deviation settings (same as S6 Fig), genes have been separated by their level of expression as in Figs 2F, 3C and 3D. The lowly expressed genes are slightly (but not significantly) better recovered when including a deviation (irrespective of its value), while middle and high expressed genes better recovered. Relative to Results subsection *Application to FANTOM5 data*.
(PDF)

**S8 Fig. Effect of varying likelihood constraints on the recovery of known regulatory relations.** For each deviation (same as S6 Fig): number of relations which present in the different relations databases (Cytreg, HTRI, Signor, Tfacts or Trrust) or at least in one (ALL). The proportion of the networks recovered is between 0.4 and 0.85% with an average of 0.56%, this number slightly lowers with a relaxed deviation but with no significance (0.6 at $\delta$ = 0 and 0.51% at 2). Relative to Results subsection *Application to FANTOM5 data*.
(PDF)

**S9 Fig. Effect of likelihood constraints deviation on the recovery of known directed regulatory relations (activation or inhibition).** Coherence of signs for the relations inferred by *Regulus* found in the Trrust and Signor databases. Out of the relations found in Trrust 42 to 63% are either unsigned or signed both ways in different contexts. Relaxing the likelihood constraints slightly reduces the quality of relation signs prediction by *Regulus*. Relative to Results subsection *Application to FANTOM5 data*.
(PDF)

**S10 Fig. Comparison of circuits between *Regulatory Circuits* and *Regulus*.** (a-c) Distribution of the number of TFs potentially regulating a gene for (a) *Regulatory Circuits* circuits, uniquely for the reduced datasets used in Fig 2, (b) *Regulus* circuits before applying the global consistency rules, (c) *Regulus* final circuits. All data presented here include non-expressed genes, explaining the high value for x = 0. (e-f) Percentage of genes from *Roadmap Epigenomics* RNA-seq datasets related to the cell populations found in circuits inferred by *Regulatory Circuits* (e) and *Regulus* (f) according to their expression level. The datasets are the ones presented in Results subsection *Application to FANTOM5 data* and Fig 2F. Relative to Fig 3 and to Results subsection *Comparison between Regulus and Regulatory Circuits*.
(PDF)

**S1 Table. Review of current circuit inference methods.** a = activity, a(t) time series of the activity, * optional, BS = TF binding site, None = description of the algorithm without

implementation. Most methods (11/15) use time series of gene expressions as the only input data, and therefore do not take the regulatory regions activity into account. Few use information about TFs binding sites or regulatory regions (among them, we identified *Regulatory Circuits* [10]) and only one([34]) checks whether the candidate TFs are expressed. The resulting circuits may then contain relations which are not consistent with the biological situation. Most methods also produce circuits with weighted edges, based on statistical or probabilistic analyses requiring large datasets acquired at several time points, which is a strong limitation to their application to human data. Indeed, many of these methods have only been tested on *Escherichia Coli* expression data and are limited to small subset of genes, raising the question of their scalability and application to human settings. Finally, we noticed that only the two most recent methods [35, 36] predict the activator or inhibitor role of the inferred regulations to generate signed circuits, but they ignore both TFs expression levels and binding site accessibility. The closest method to what we aim for is *Regulatory Circuits*, but it still shows some design and reproducibility issues, as shown in the main text. Relative to the Introduction.
(PDF)

**S2 Table. Recovery of known regulatory relations from *Regulatory Circuits* networks.** Number of relations present in the different relations databases (Cytreg, HTRI, Signor, Tfacts or Trrust) present in the computed networks. Overall, *Regulatory Circuits* is enriched in relations found in databases, with significant enrichment obtained in 19 out of 20 cases (p-values ranging from 4.6e-3 to 1.7e-290). Relative to Fig 3.
(PDF)

**S3 Table. Descriptive statistics on gene expression patterns and TF-gene relations obtained by applying *Regulus* to human B cell subsets.** Data were processed with a restrictive set of TF binding sites, filtered to have a strictly positive score as found in the supplementary data of [10]. ND: not determined, as all consistent relations involving constantly expressed genes also involve TFs and regions with constant activities; it is therefore not possible to qualify these relations as activation or inhibition. *: percentage taking into account the undetermined relations of the *5555* pattern. Relative to Fig 4.
(PDF)

**S1 Data. Document in xls format containing, in separate sheets, the underlying numerical data for Figs 2C, 2D, 4B and 4D.**
(XLS)

## Acknowledgments

We acknowledge the GenOuest bioinformatics core facility for providing the computing infrastructure (https://www.genouest.org).

## Author Contributions

**Conceptualization:** Marine Louarn, Olivier Dameron, Anne Siegel, Fabrice Chatonnet.

**Funding acquisition:** Thierry Fest.

**Methodology:** Marine Louarn, Ève Barré, Olivier Dameron, Anne Siegel, Fabrice Chatonnet.

**Software:** Marine Louarn, Guillaume Collet, Ève Barré.

**Supervision:** Thierry Fest, Olivier Dameron, Anne Siegel, Fabrice Chatonnet.

**Validation:** Marine Louarn.

**Writing – original draft:** Marine Louarn, Fabrice Chatonnet.

**Writing – review & editing:** Marine Louarn, Olivier Dameron, Anne Siegel, Fabrice Chatonnet.

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
