## [Decision Letter · Decision Letter 0]

30 Mar 2023

Dear Dr Chatonnet,

Thank you very much for submitting your manuscript "Regulus infers signed and process-based regulatory circuits from few samples" for consideration at PLOS Computational Biology.

As with all papers reviewed by the journal, your manuscript was reviewed by members of the editorial board and by several independent reviewers. In light of the reviews (below this email), we would like to invite the resubmission of a significantly-revised version that takes into account the reviewers' comments.

We cannot make any decision about publication until we have seen the revised manuscript and your response to the reviewers' comments. Your revised manuscript is also likely to be sent to reviewers for further evaluation.

Sincerely,

Ilya Ioshikhes

Section Editor

PLOS Computational Biology

Lucy Houghton

Staff

PLOS Computational Biology

Reviewer's Responses to Questions

**Comments to the Authors:**

Reviewer #1: The manuscript presents a method (“Regulus”) to find regulators of genes and classify these as activators of inhibitors. Using datasets as inputs. Authors claim the method is suitable for experiments with “few” samples, thus filling a gap within the literature.

The problem tackled sounds important, and the solution suggested seems to be elegant, but I cannot see the results presented in an understandable way. I would encourage authors to improve the manuscript. May the following comments be useful:

1) Figures have very low quality. I cannot see but just a portion of the info that is in there. Thus I am not able to judge the results properly.

2) The solution is not benchmarked against existing tools (or itself). I find table S1 insufficient. Authors claim the other methos are not “easily implemented”, which needs more information. How does the tool perform using the same problem and same measurements compared against others? If using “few samples” is the advantage… how many is “few”? Is there a limit? Which one?

3) Thorough the manuscript authors say to be using “logical consistency check translated from biological knowledge”, “a score which was estimated according to experts” and similar statements to refer to key areas of the method. What do authors mean? Can authors provide a more specific explanation?

4) “By testing Regulus on unpublished biological datasets”. Why using unpublished datasets? How can readers evaluate performance?

5) The species is not mentioned till well into the manuscript. I suggest mentioning what is the species (what cells) being used up front. Also, the fact that the ambition is to target “patho-physiological” conditions and “B cells” should be stated up front. All these are important pieces of info to frame the problem at stake.

6) I would suggest differentiating between the implementation of the tool (Semantic Web, pipeline details, etc.) and the results / performance of the tool. So far all this info is mixed and the resulting manuscript is difficult to read.

7) If “entity pattern” is something important (which looks like), then highlight it earlier on. If it is not, I would suggest moving this info to Methods. It is not clear what a pattern is.

8) When talking about “computing the distances between them”. What do authors mean? Base-pair distances I guess?

9) Authors state they are using “four” cell populations from the FANTOM5 dataset. Which ones? Are these NBC, IgM+…? The names of these is a couple pages below. Also, why these ones? How do other tools perform on these specific ones? It looks like table S1 is on a “B dataset” (where is this?)

10) Lines 378 to 387 outline 4 advantages of “Regulus”. I don’t think these advantages are obvious from the text. For instance, take advantage #1 (“under exploitation of the regulatory context”): how under exploited? Any quantification?

Reviewer #2: In this manuscript, the authors report the design of a transcription regulatory inferencing tool, Regulus, which can unravel TF-gene interactions such as activation or inhibition by taking into consideration certain epigenetic information, even with limited data. The idea of using Semantic Web technologies for integrating gene expression data with epigenetic data is neat, and makes it possible to infer context dependent TF-gene interactions making Regulus a powerful tool. The premise is quite interesting, but there are a few more clarifications that are needed about the method, to further strengthen the manuscript.

Major comments:

1. Line 106: "Our circuit inference method is based on the common assumption that genes sharing a common expression dynamics are regulated by a common set of regulators." According to the inferences drawn by Regulus, are the regulatory interactions (activation/inhibition of the genes by the common set of regulators) also same or different for genes sharing common expression dynamics?

2. Did the authors encounter an example of regulatory inference derived using Regulus that varied with difference in contexts? Highlighting such an example and if possible the implications of wrongly inferring the interactions could be a good addition to the paper.

3. Although Regulus helps to zero-in on the most relevant TFs regulating a gene, it does not appear to infer the dynamics of their combined effect. For instance, it does not provide information whether the TFs follow AND, OR or competitive binding. Is this correct -- this should be discussed.

4. The authors claim that Regulus is able to infer the TF-gene interactions with limited data over similar cell types. However, it requires both RNA-seq and ATAC-seq data, which may not be available in many scenarios. Could the authors comment on this?

5. In terms of comparisons, there is only a set of comparisons with one tool, Regulatory Circuits. While it is understood that it is the closest method of circuit inference, I am wondering if some predictions can be compared with other databases/tools/studies. This seems to be a slight weakness in this manuscript. Is there a way to more quantitatively compare the results?

6. Line 459: "All genomic coordinates are given according to the hg19 human reference genome." Why not hg38, which has become the standard for many years now? This will impact the adoption and longevity of Regulus.

7. At times, I found the paper a bit difficult to read, requiring a bit of back-and-forth between various sections. While it's mostly unavoidable, I am wondering if a toy example, to go with the workflow diagram, will be possible, and can elucidate the methodology better.

Minor comments:

1. The resolution of ‘RDF data model structure’ inset in Fig 1 should be improved.

2. Line 355: 19/20 cases, p-values - could these be supplied in a Supplementary Table?

3. In the label for Fig 3. the Pie chart is marked as B and Table as C but in the Fig, it's the opposite.

4. Line 484: The number of bins was chosen as four. While this is justified in the text, I wonder if the effect of number of bins has a clear-cut effect on the predictions/performance. Also, it may be interesting to look at the (extreme) Boolean case, as Boolean networks are routinely used to (even if simplistically) model transcriptional regulatory networks.

5. Line 598: less than symbol is not in math mode and hasn't reproduced correctly.

**Have the authors made all data and (if applicable) computational code underlying the findings in their manuscript fully available?**

Reviewer #1: **No: **One of my comments: 4) “By testing Regulus on unpublished biological datasets”. Why using unpublished datasets? How can readers evaluate performance?

Reviewer #2: Yes

PLOS authors have the option to publish the peer review history of their article (what does this mean?). If published, this will include your full peer review and any attached files.

Reviewer #1: No

Reviewer #2: No
---

## [Decision Letter · Decision Letter 1]

23 Nov 2023

Dear Dr Chatonnet,

Thank you very much for submitting your manuscript "Regulus infers signed regulatory relations from few samples’ information using discretization and likelihood constraints" for consideration at PLOS Computational Biology. As with all papers reviewed by the journal, your manuscript was reviewed by members of the editorial board and by several independent reviewers. The reviewers appreciated the attention to an important topic. Based on the reviews, we are likely to accept this manuscript for publication, providing that you modify the manuscript according to the recommendations of reviewer 2.

Sincerely,

Ilya Ioshikhes

Section Editor

PLOS Computational Biology

Lucy Houghton

%CORR_ED_EDITOR_ROLE%

PLOS Computational Biology

Reviewer's Responses to Questions

**Comments to the Authors:**

Reviewer #1: I went through the revision, and the rebuttal letter. I consider my concerns addressed, so I recommend publication as is.

Reviewer #2: All my original comments have been satisfactorily addressed.

Just one additional point to consider: Could the authors specify what is the main bottleneck when applying Regulus to a dataset that has a larger number of samples? Since Regulus incorporates a lot of contextual information in arriving at its inferences, if the method could scale for large datasets, it could potentially be used to validate the results obtained by other inference methods which are more data-centric and are less biology-informed in their approaches.

**Have the authors made all data and (if applicable) computational code underlying the findings in their manuscript fully available?**

Reviewer #1: Yes

Reviewer #2: Yes

PLOS authors have the option to publish the peer review history of their article (what does this mean?). If published, this will include your full peer review and any attached files.

Reviewer #1: No

Reviewer #2: No

Figure Files:

Data Requirements:

Reproducibility:

References:

---

## [Editor Report · Decision Letter 2]

8 Jan 2024

Dear Dr Chatonnet,

We are pleased to inform you that your manuscript 'Regulus infers signed regulatory relations from few samples’ information using discretization and likelihood constraints' has been provisionally accepted for publication in PLOS Computational Biology.

Best regards,

Ilya Ioshikhes

Section Editor

PLOS Computational Biology

Lucy Houghton

%CORR_ED_EDITOR_ROLE%

PLOS Computational Biology

---

## [Editor Report · Acceptance letter]

18 Jan 2024

PCOMPBIOL-D-22-01850R2 

*Regulus* infers signed regulatory relations from few samples’ information using discretization and likelihood constraints

Dear Dr Chatonnet,

I am pleased to inform you that your manuscript has been formally accepted for publication in PLOS Computational Biology. Your manuscript is now with our production department and you will be notified of the publication date in due course.

With kind regards,

Zsuzsanna Gémesi
